Sirtuin 1 alleviates alcoholic liver disease by inhibiting HMGB1 acetylation and translocation

Fu Juan fujuan320@126.com 1
Deng Wei 2
Ge Jun 1
Fu Shengqi 1
Li Panpan 1
Wu Huazhi 1
Wang Jiao 1
Gao Yi 1
Gao Hui 1
Wu Tao 1
1 Department of Infectious Diseases, Hainan General Hospital/Hainan Affiliated Hospital of Hainan Medical University , Haikou , China
2 Department of Oral and Maxillofacial Surgery, Hainan General Hospital/Hainan Affiliated Hospital of Hainan Medical University , Haikou , China
Guan Fanglin
Electronic publication date: 2023 Nov 27
Publication date: 2023
Volume: 11
Electronic Location ID: e16480
Received 2023 Jul 10; Accepted 2023 Oct 26
Copyright: ©2023 Fu et al.
Copyright year: 2023
Copyright holder: Fu et al.
License: This is an open access article distributed under the terms of the Creative Commons Attribution License, which permits unrestricted use, distribution, reproduction and adaptation in any medium and for any purpose provided that it is properly attributed. For attribution, the original author(s), title, publication source (PeerJ) and either DOI or URL of the article must be cited.
License URL: https://creativecommons.org/licenses/by/4.0/

Keywords: SIRT1, HMGB1, Alcoholic liver disease, Acetylation, Translocation

Funding: Hainan Provincial Natural Science Foundation of China 820MS136 This study was supported by the Hainan Provincial Natural Science Foundation of China (No. 820MS136). The funders had no role in study design, data collection and analysis, decision to publish, or preparation of the manuscript.

==============================
Background

Alcoholic liver disease (ALD) encompasses a spectrum of liver disorders resulting from prolonged alcohol consumption and is influenced by factors such as oxidative stress, inflammation, and apoptosis. High Mobility Group Box 1 (HMGB1) plays a pivotal role in ALD due to its involvement in inflammation and immune responses. Another key factor, Sirtuin 1 (SIRT1), an NAD+-dependent deacetylase, is known for its roles in cellular stress responses and metabolic regulation. Despite individual studies on HMGB1 and SIRT1 in ALD, their specific molecular interactions and combined effects on disease advancement remain incompletely understood.

Methods

Alcohol-induced liver injury (ALI) models were established using HepG2 cells and male C57BL/6 mice. HMGB1 and SIRT1 expressions were assessed at the mRNA and protein levels usingreverse transcription-quantitative polymerase chain reaction, western blot, and immunofluorescence staining. The physical interaction between HMGB1 and SIRT1 was investigated using co-immunoprecipitation and immunofluorescence co-expression analyses. Cellular viability was evaluated using the CCK-8 assay.

Results

In patients with clinical ALI, HMGB1 mRNA levels were elevated, while SIRT1 expression was reduced, indicating a negative correlation between the two. ALI models were successfully established in cells and mice, as evidenced by increased markers of cellular and liver damage. HMGB1 acetylation and translocation were observed in both ALI cells and mouse models. Treatment with the SIRT1 agonist, SRT1720, reversed the upregulation of HMGB1 acetylation, nuclear translocation, and release in the ethyl alcohol (EtOH) group. Furthermore, SIRT1 significantly attenuated ALI. Importantly, in vivo binding was confirmed between SIRT1 and HMGB1.

Conclusions

SIRT1 alleviates HMGB1 acetylation and translocation, thereby ameliorating ALI.

Introduction

Attributed to prolonged alcohol consumption, alcoholic liver disease (ALD) is a chronic ailment with long-term implications, progressing over time to conditions such as alcoholic hepatitis, alcoholic fatty liver, alcoholic cirrhosis, and even liver cancer (Kourkoumpetis & Sood, 2019). Key contributors to its pathogenesis include oxidative stress, inflammation, and stem cell apoptosis (Namachivayam & Valsala Gopalakrishnan, 2021). The liver, pivotal in ethanol metabolism, is significantly impacted by alcohol intake (Liu, Tsai & Hsu, 2021). With changing lifestyles and increased alcohol availability, alcohol-induced liver injury (ALI) incidence has been rising, posing a significant public health threat. Consequently, intensifying ALI research is necessary to discover safe and effective treatment modalities.

High Mobility Group Box 1 (HMGB1), a versatile and ubiquitous non-histone protein found in nearly all eukaryotic cells (Yang, Wang & Andersson, 2020), undergoes posttranslational modifications, such as acetylation, phosphorylation, and n-glycosylation, to modulate its structure and biological functions (Lundbäck et al., 2016; Vergoten & Bailly, 2020; Wang et al., 2022). Studies indicate elevated levels of acetylated HMGB1 in patients with H2O2-induced liver injury (Kwak et al., 2019). Neutralizing HMGB1 with antibodies significantly improves liver damage and survival in a D-galactosamine-6-induced liver failure model (Lundbäck et al., 2016), highlighting its potential as a liver biomarker for early diagnosis and disease monitoring. Recent reports emphasize the importance of sirtuins in regulating metabolic processes relevant to ALD (Kim et al., 2019; Ma et al., 2019). Ethanol exposure reduces SIRT1 gene and protein expressions (You et al., 2015). Decreased SIRT1 levels play a pivotal role in ALD regulation through alterations in acetylation patterns of various target molecules, including histones, transcription regulatory factors, and co-regulatory factors (Chang & Guarente, 2014; Ren et al., 2020; Schug & Li, 2011). Notably, SIRT1-mediated HMGB1 deacetylation mitigates acute kidney injury associated with sepsis (Wei et al., 2019). Moreover, HMGB1 plays a crucial role in alcoholic steatohepatitis by inducing endoplasmic reticulum (ER) stress in hepatocytes, triggering intensified inflammatory responses and apoptosis. The present study highlights the significance of HMGB1 in ALD development and proposes inhibiting HMGB1 as a potential therapeutic avenue (Gan et al., 2014). However, the relationship between SIRT1 and HMGB1 regulation in ALD remains unexplored.

HepG2 cells and male C57BL/6 mice were used in this study to establish in vitro (IVT) and in vivo (IVV) ALI models. Molecular assays assessed HMGB1 and SIRT1 expressions and verified their interactions. Subsequent functional experiments explored the regulatory mechanism of SIRT1 on HMGB1 in ALD pathogenesis, offering novel insights into ALD pathogenesis and a theoretical basis for developing new clinical interventions.

Material and Methods

Clinical samples

Liver tissue samples were obtained by clinicians from patients with ALI, and 20 liver tissue samples were obtained from healthy individuals collected at Hainan General Hospital. This study was approved by the Hainan General Hospital Ethics Committee ([2022]318), and all patients provided written informed consent. The human samples used in this study adhere to the standards of the Declaration of Helsinki.

siRNAs and reagents

Human SIRT1-specific siRNA (siSIRT1: 5′-CAGACAGUGCAGUAUUCACTT-3′) and control siRNA (#4427037) were sourced from Thermo Fisher Scientific (Waltham, MA, USA). Lipofectamine RNAiMAX (#13778150, Thermo Fisher Scientific, Waltham, MA, USA) was used for siRNA transfection following the manufacturer’s instructions. In addition, alcohol (CAS No. (64-17-5), Sinopharm Chemical Reagent Co. Ltd., Shanghai, China), SRT1720 (Selleckchem, Houston, TX, USA), EX527 (Selleckchem), anti-SIRT1 (1:1,000, ab110304, Abcam, Cambridge, MA, USA), anti-GAPDH (1:1,000, Proteintech, Rosemont, IL, USA), anti- HMGB1 (1:1,000, ab79823, Abcam), and anti-histone (1:1,000, ab1791, Abcam) were utilized in the study.

Cell culture and treatment

HepG2 human hepatocyte cells from American Type Culture Collection (Manassas, Virginia, USA) were cultured in Dulbecco’s Modified Eagle Medium (Gibco, Waltham, MA, USA) supplemented with 1% penicillin and streptomycin (Invitrogen, Carlsbad, CA, USA) and 10% fetal bovine serum (Gibco, USA) at 37 °C in a humidified incubator with 5% CO2. HepG2 cells were exposed to ethanol concentrations of 1%, 2%, 3%, and 4% to establish an ALI cell model.

Nuclear extraction

A nuclear and cytoplasmic protein extraction kit (#78833, Pierce Biotechnology, Rockford, IL, USA) was used to extract nuclear and cytoplasmic proteins according to the manufacturer’s instructions. Briefly, 2 × 105 cells were harvested with trypsin-EDTA, centrifuged at 500 × g for 5 min, and resuspended in PBS. After another centrifugation at 500 × g for 2 min, the supernatant was discarded, and the cell pellet was incubated with ice-cold CER I and CER II for 1 min. For western blot experiments, cytoplasmic and nuclear extracts were collected after centrifugation at 16,000 × g.

CCK-8

HepG2 cells in the logarithmic growth phase were trypsinized and seeded into 96-well culture plates at a density of 4 × 103 (200 µL)/well. After 24 h of treatment, 10 µL of CCK-8 solution (Dojindo Molecular Technologies, Inc., Kumamoto, Japan) was added to each well. After a 2-h incubation at 37 °C, absorbance at 450 nm was measured using a microplate reader (SpectraMax M2, Molecular Devices, Silicon Valley, CA, USA).

Quantitative real-time polymerase chain reaction (RT-qPCR)

Total RNA from HepG2 cells or liver tissues was extracted using Trizol reagent (Invitrogen, USA). RNA concentration and purity were assessed using a NanoDrop2000 spectrophotometer measuring A260 and A260/280 ratios, respectively. PrimeScript™ RT Master Mix (Takara Biomedical Technology Co., Ltd., Beijing, China) was used to reverse-transcribe RNA into cDNA. Quantitative real-time polymerase chain reaction (RT-qPCR) was performed using the Light Cycler 480 system, which underwent amplification of 40 cycles with denaturation at 94 °C for 5 s and annealing/extension at +60 °C for 30 s. The Ct value difference represented gene transcription differences, and 2−ΔΔCt indicated the relative gene expression. Primers used are listed in Table 1.

Table 1 Primers used for RT-qPCR.

Genes	Forward (5′-3′)	Reverse (5′-3′)	
SIRT1	TCATTCTGACTGTGATGACGA	CTGCCACAGTGTCATATCCAA	
HMGB1	AGAAGTGCTCAGAGAGGTGGA	CCTTTGGGAGGGATATAGGTT	
GAPDH	CCCATCACCATCTTCCAGGAG	GTTGTCATGGATGACCTTGGC	

Western blot and co-immunoprecipitation

Total protein from HepG2 cells and liver tissues was extracted using RIPA lysate (Pierce; Thermo Fisher Scientific, Inc.) supplemented with protease and phosphatase inhibitors. Protein concentration was determined using a BCA kit. For co-immunoprecipitation, lysates containing 500 µg of total protein and 2 µg of specific antibodies were incubated at 4 °C for 18 h. Subsequently, 50 µL of protein G agarose beads were added, followed by a 3-h incubation. After five washes using cracking buffer, the precipitated protein was suspended in 30 µL of sodium dodecyl sulfate (SDS) sample buffer and boiled at 95 °C for 10 min.

After loading 30 µg of protein, 10% SDS polyacrylamide gel electrophoresis was performed, followed by the transfer of proteins onto a polyvinylidene fluoride (PVDF) membrane (Nanjing Institute of Bioengineering, Nanjing, China). Subsequently, the PVDF membrane was immersed in 5% skim milk for 2 h, and the primary antibody was then applied to incubate the membrane overnight at 4 °C. After a 2-h washing with TBS at room temperature, the membrane was incubated with horseradish peroxidase-labeled goat anti-rabbit or mouse IgG (1:5000, sc-516102/sc-2357; Santa Cruz Biotechnology, Inc. Dallas, TX, USA). Target bands were visualized using an ECL kit (Biyuntian Biotechnology Co., Ltd., Shanghai, China), and band intensities were quantified using Image J software.

Immunofluorescence staining

HepG2 cells were fixed with 4% formaldehyde for 30 min at room temperature, permeabilized with 0.1% Triton X-100 at 4 °C for 10 min, and blocked with 2% bovine serum albumin in PBS for 1 h at room temperature. Immunofluorescence was conducted using an anti-HMGB1 (L7543, Sigma, Burlington, MA, USA) or anti-SIRT1 antibody, followed by Alexa Fluor 488 conjugate immunoglobulin and DAPI (Sigma). Fluorescence signals were examined using an Olympus Fluoview 1000 confocal microscope (Olympus Corp, Tokyo, Japan).

Mouse studies

Twelve adult male C57BL/6 mice (5 weeks old, 21–23 g) were obtained from Guangdong Medical Laboratory Animal Center (SCXK (Guangdong, China) 2022-0002). Mice were housed in a specific pathogen-free environment in an IVC box (H6, Suzhou Suhang Technology Equipment Co., Ltd., Suzhou, China) with three mice per box, maintained at 40%–70% humidity, 20 °C–26 °C temperature, and a 12-h dark-light cycle. Mice were divided into two groups: Control and ALD model groups (n = 6 mice per group). The Gao-Binge model was employed to induce ALD in mice (Qian, 2022). After adapting to Lieber-DeCarli’s control diet for five days, the ALD model group was fed the Lieber-DeCarli liquid diet containing 5% alcohol (SPF grade diet, Jiangsu Synergistic Bioengineering Co., Ltd.). Mouse weights were measured daily in both groups. On the 16th day, alcohol intragastric administration was performed, and 9 h later, mice were euthanized by cervical dislocation. Liver tissues were collected, and blood was obtained from the eyeballs to determine the liver-body weight ratio. This study was approved by the Ethical Committee of Hainan General Hospital ([2022]318).

Hematoxylin and eosin and oil red

Liver tissue sections were dewaxed, hydrated using xylene and alcohol (70%, 90%, and 100%, v/v), and subjected to hematoxylin and eosin staining for visualization. An optical microscope was used to observe the liver tissues. Oil Red O staining was employed to identify tissue lipidosis.

Enzyme-linked immunosorbent assay (ELISA)

In the double antibody clip enzyme-linked immunosorbent assay (ELISA), mouse monoclonal antibodies were affixed to enzyme-labeled plates. TNF-α (RayBiotech, Norcross, GA, USA), IL-6 (RayBiotech), or IL-1β (RayBiotech) was incubated with the monoclonal antibody, and unbound components were subsequently washed away. Biotinylated anti-mouse TNF-α, IL-6, or IL-1β antibodies were introduced, followed by the addition of horseradish peroxidase-labeled avidin, forming specific binding with biotin-avidin. The combination of anti-mouse TNF-α, IL-6, or IL-1β antibodies with enzyme-labeled antibodies, along with horseradish peroxidase binding to the monoclonal antibody, initiated a color change from colorless to blue and then yellow upon adding a termination solution. The optical density (OD) value was then measured at 450 nm, allowing for the determination of TNF-α, IL-6, or IL-1β concentrations in samples using a standard curve.

Detection of biochemical indexes

The malondialdehyde (MDA), aspartate aminotransferase (AST), and alanine aminotransferase (ALT) test kits used were purchased from Nanjing Jiancheng Bioengineering Institute. In addition, triglyceride (TG) and total cholesterol (TC) kits were sourced from Beijing Beihua Kangtai Clinical Reagent Co., Ltd (Beijing, China).

Statistical analysis

Data was presented as mean ± standard deviation. Statistical significance between groups was assessed using analysis of variance followed by Dunnett’s t-test, conducted with GraphPad Prism 9.0 (Graph-Pad Software Inc., La Jolla, CA, USA). p <0.05 was considered statistically significant.

Results

Negative correlation between HMGB1 and SIRT1 expressions in ALD

In the initial investigation, HMGB1 and SIRT1 expressions were examined in patients with clinical ALD. RT-qPCR analysis revealed an upregulation of HMGB1 mRNA expression (Fig. 1A), while SIRT1 expression showed downregulation (Fig. 1B) in alcoholic liver tissues compared to normal liver tissues. Pearson correlation analysis demonstrated a negative correlation between HMGB1 and SIRT1 expressions (Fig. 1C). These findings indicated the significant roles of HMGB1 and SIRT1 in ALD pathogenesis.

Figure 1 A negative correlation was observed between HMGB1 and SIRT1 levels in patients with alcoholic liver disease (ALD).

(A) Elevated HMGB1 mRNA levels were observed in patients with alcoholic livers, (n = 20). (B) Reduced SIRT1 mRNA levels were observed in patients with alcoholic livers, (n = 20). (C) A negative relationship was observed between HMGB1 and SIRT1 in patients with alcoholic livers. **p <0.01, ***p <0.001. Pearson correlation analysis was performed.

HMGB1 acetylation and translocation ethanol-treated HepG2 cells

To establish the ALI cell model, HepG2 cells were exposed to various ethanol concentrations. Cell viability decreased to approximately 95%, 90%, 70%, and 70% when treated with 1%, 2%, 3%, and 4% ethanol, respectively (Fig. 2A). Consequently, 3% ethanol was selected for subsequent analysis. HepG2 cells exposed to 3% ethanol exhibited a significant increase in reactive oxygen species (ROS) production compared to the control group (Fig. 2B), along with elevated MDA levels (Fig. 2C). Moreover, these ethanol-exposed cells showed a significant increase in ALT and AST levels compared to the control group (Figs. 2D and 2E), confirming the successful establishment of the ALI cell model. Given the observed upregulation of HMGB1 mRNA in patients with clinical ALD, HMGB1 expression was examined in 3% ethanol-exposed HepG2 cells using immunofluorescence staining. The EtOH group showed a significant increase in HMGB1 expression compared to the control group (Fig. 2F). In this study, we also explored the localization and translocation mechanism of HMGB1, revealing that ethanol treatment decreased HMGB1 protein expression in the nucleus but increased in the cytoplasm of HepG2 cells (Fig. 2G). The results confirming the purity of the isolated nuclear and cytosol fractions can be found in Fig. S1. Previous studies highlighted the significance of post-translational modifications, including acetylation, in HMGB1’s nuclear-to-cytoplasmic translocation. Notably, HMGB1 acetylation showed a significant increase in the EtOH group (Fig. 2G), suggesting a potential role for acetylation in facilitating its translocation.

Figure 2 HMGB1 exhibited acetylation and translocation from the nucleus to the cytoplasm in HepG2 cells following ethanol treatment.

(A) HepG2 cell viability was inhibited by increasing ethanol concentrations (1%–4%), (n = 3). (B) ROS production was increased with 3% ethanol in HepG2 cells, (n = 3). (C) MDA content was elevated in HepG2 cells exposed to 3% ethanol, (n = 3). (D) ALT levels increased after 3% ethanol treatment in HepG2 cells, (n = 3). (E) AST levels increased in HepG2 cells after 3% ethanol treatment, (n = 3). (F) Immunofluorescence analysis confirmed elevated HMGB1 levels in HepG2 cells after 3% ethanol treatment, (n = 3). (G) Western blot analysis revealed HMGB1 acetylation and translocation from the nucleus to the cytoplasm in HepG2 cells after ethanol treatment. *p <0.05, **p <0.01, ***p <0.001, ****p <0.0001.

SIRT1 regulation of HMGB1 acetylation and translocation

Previous research highlighted the regulatory role of SIRT1 in HMGB1 under various disease conditions, such as sepsis-associated acute kidney injury. To investigate this in the context of ALI, the selective SIRT1 activator SRT1720, the inhibitor Ex527, and SIRT1 siRNA were employed in co-incubation with ethanol-exposed HepG2 cells. Western blot analysis revealed that Ex527 further enhanced protein acetylation and cytoplasmic HMGB1 levels induced by ethanol while reducing nuclear HMGB1 protein expression (Fig. 3A). In contrast, SRT1720 counteracted the impact of ethanol on HMGB1 acetylation and translocation within HepG2 cells (Fig. 3A). Similarly, SIRT1 siRNA had a comparable effect to the SIRT1 inhibitor Ex527 on ethanol-exposed HepG2 cells (Fig. 3A), suggesting a functional interaction between SIRT1 and HMGB1. The results confirming the purity of the isolated nuclear and cytosol fractions can be found in Fig. S2. Using co-immunoprecipitation experiments, the physical binding between SIRT1 and HMGB1 was validated (Fig. 3B). Immunofluorescence staining further demonstrated the co-localization of SIRT1 and HMGB1, supporting their interaction (Fig. 3C).

Figure 3 SIRT1 suppressed HMGB1 acetylation and nucleus-to-cytoplasm translocation.

(A) HepG2 cells were pretreated with SIRT1 siRNA, NC siRNA, SRT1720, and EX527 before ethanol exposure. Western blot analysis assessed SIRT1, Nucleus-HMGB1, Cytoplasm-HMGB1, and Acetyl-HMGB1 protein expressions. (B) Co-immunoprecipitation experiment confirmed the physical interaction between SIRT1 and HMGB1. (C) Co-localization of SIRT1 and HMGB1 in HepG2 cells was observed, (n = 3).

Protective role of SIRT1 against ALI

To investigate the protective effects of SIRT1 against ALI in HepG2 cells, the SIRT1 activator SRT1720, the inhibitor Ex527, and SIRT1 siRNA were used in co-incubation with ethanol-exposed HepG2 cells. Alcohol treatment reduced cell viability, which was exacerbated by EX527 and SIRT1 siRNA, while restored by SRT1720 (Fig. 4A). Alcohol-induced upregulation in ROS production and MDA were amplified by EX527 and SIRT1 siRNA but mitigated by SRT1720 in ethanol-exposed HepG2 cells (Figs. 4B and 4C). Ethanol exposure significantly increased ALT and AST levels, which were further enhanced by EX527 and SIRT1 siRNA, while SRT1720 reduced ALT and AST levels (Figs. 4D and 4E). The combined data from Figs. 3 and 4 strongly support the role of SIRT1 in attenuating ALI by regulating HMGB1 acetylation.

Figure 4 SIRT1 attenuated ALI.

(A) Both EX527 and siSIRT1 further reduced HepG2 cell viability after ethanol treatment, (n = 3). (B) Ethanol-induced ROS production was upregulated by both EX527 and siSIRT1 in HepG2 cells, (n = 3). (C) MDA levels were increased by both EX527 and siSIRT1 in HepG2 cells after ethanol treatment, (n = 3). (D) ALT levels were elevated by both EX527 and siSIRT1 in HepG2 cells after ethanol treatment, (n = 3). (E) Enhanced AST levels were observed in HepG2 cells treated with EX527 and siSIRT1 after ethanol treatment, (n = 3). *p <0.05, **p <0.01, ***p <0.001, ****p <0.0001.

SIRT1 and HMGB1 expressions in ethanol-treated C57BL/6 mice

To confirm the relationship between SIRT1 and HMGB1, an ALI mice model was established. Histological examination of liver sections using H&E and Oil Red O staining revealed normal morphology in the control group but evident liver pathogenesis in the EtOH group (Fig. 5A). The liver-to-body weight ratio significantly increased in ethanol-treated mice (Fig. 5B). As anticipated, mice subjected to ethanol feeding showed significant upregulation in TC and TG serum levels compared to the control group (Figs. 5C and 5D), along with increased ALT and AST levels (Figs. 5E and 5F). Consistent with findings in patients with chronic alcoholism, liver tissues of ethanol-treated mice exhibited increased levels of the inflammatory mediators IL-1β, IL-6, and TNF-α (Figs. 5G–5I). Furthermore, the ethanol-treated mice showed reduced SIRT1 expression, accompanied by increased HMGB1 acetylation and translocation (Fig. 5J). The results confirming the purity of the isolated nuclear and cytosol fractions can be found in Fig. S3. These results confirmed the successful establishment of the ALI model.

Figure 5 Low SIRT1 expression and HMGB1 acetylation and translocation were observed in C57BL/6 mice following ethanol treatment.

(A) Liver sections stained with H&E and Oil Red O were acquired from the control and EtoH groups. (B) Liver-to-body ratio in the control group and C57BL/6 mice after ethanol treatment, (n = 6). (C–F) Serum levels of TC, TG, ALT, and AST, (n = 6). (G-I) Serum levels of IL-1 β, IL-6, and TNF- α, (n = 6). (H) Protein levels of SIRT1, Nucleus-HMGB1, Cytoplasm-HMGB1, and Acetyl-HMGB1. Data are shown as mean ± SD (n = 6). *p <0.05, **p <0.01, ***p <0.001, ****p <0.0001.

Discussion

Previous research effectively linked ALD to various factors, including oxidative stress, inflammatory reaction, lipid metabolism disorder, mitochondrial dysfunction, and endotoxin exposure—all stemming from excessive alcohol consumption (Dukić et al., 2023). Chronic heavy alcohol intake can result in liver lipid metabolism dysfunction, microtubule damage within liver cells, and the intracellular accumulation of fat and secretory proteins, leading to an elevated liver index (Wang et al., 2022). This liver injury leads to increased liver cell membrane permeability, releasing enzymes such as AST and ALT into the bloodstream, subsequently elevating blood AST and ALT levels (Malnick et al., 2022). Ethanol metabolism generates acetaldehyde, disrupting the equilibrium between the body’s oxidative and peroxide systems, resulting in oxidative stress. This imbalance contributes to increased levels of cytotoxic oxidative product MDA in liver tissues, increased ROS content, compromised antioxidant capacity, reduced free-radical scavenging ability, and lipid peroxidation of liver cell membranes, collectively causing liver cell damage (Mai et al., 2022). In the present study, the EtOH group showed significantly increased ALT and AST levels in HepG2 cells compared to the control group, indicating liver cell damage. Simultaneously, increased MDA and ROS levels were observed in the EtOH group, signifying oxidative stress-induced damage in the samples. These findings were further validated in mouse models and supported by histochemical experiments, thereby confirming the successful establishment of both the ALI cell and mouse models.

Recent studies have established the increased HMGB1 levels in ALD, contributing to liver damage (Bukong et al., 2018). Alcohol intake leads to increased HMGB1 acetylation and translocation from the nucleus to the cytoplasm (Fuster & Samet, 2018). In fibrotic liver disease, HMGB1 is significantly elevated and plays a central role in hepatic fibrosis (Ge et al., 2018). Notably, in liver ischemia/reperfusion injury and non-ALD conditions, extracellular HMGB1 prominently contributes to inflammation in liver injury (Zhang et al., 2013). HMGB1 functions as an inflammatory cytokine, and its inhibition and transport blockage have reportedly prevented non-alcoholic steatohepatitis (Zeng et al., 2015). In addition, HMGB1 is linked to autophagy induction, promoting the activation of hematopoietic stem cells and liver fibrosis (Li et al., 2018a). Consistent with earlier research, HMGB1 showed increased expression in patients with ALD and ALD models, accompanied by its acetylation and translocation.

As a member of the NAD+-dependent deacetylase sirtuin family, SIRT1 regulates liver metabolism and physiology by producing and secreting FGF21 (Li et al., 2014). Loss of SIRT1 in bone marrow cells, including neutrophils, enhances ALI and inflammation in mice (Ren et al., 2022). In young mice, SIRT1 overexpression in the liver prevents obesity-induced liver ER stress and insulin resistance (Li et al., 2011). Hepatocyte-specific SIRT1 deficiency exacerbates ALD in both mice and humans through the detor-mTORC1 signaling pathway (Chen et al., 2018). In addition, SIRT1 depletion in hepatic stellate cells aggravates hepatic fibrosis following bile duct ligation in young mice (Li et al., 2018b). In the present study, the ALI mouse model showed reduced SIRT1 protein expression. Further inhibition of SIRT1 worsened ALI, while the SIRT1 agonist SRT1720 alleviated it. In addition, SRT1720 reversed HMGB1 acetylation and translocation in ALD. The research also confirmed a physical interaction between HMGB1 and SIRT1. These results collectively suggest that SIRT1 likely exerts protective effects against ALD by inhibiting HMGB1 acetylation and translocation in both ALD cell and mouse models. While our study provides preliminary evidence of HMGB1 and SIRT1 in ALI, the limited number of clinical samples underscores the need for more patient data to corroborate the association between HMGB1 and SIRT1 and their contributions to ALD development.

While this study provides valuable insights into the pathological features of the disease model, several limitations should be acknowledged. First, the animal model used in this study may not fully replicate all facets of human ALD, given its multifactorial nature influenced by genetics, environment, and lifestyle. Thus, caution is necessary when applying these findings to patients with ALD. Second, the study predominantly examined the pathological features at a cellular level, neglecting systemic interactions and the intricate dynamics within the entire organism. Hence, future investigations employing biological methodologies could offer a more comprehensive grasp of ALD progression and its interplay with different organs and systems.

Building upon the insights gained from this study, several promising avenues for future research can be identified. First, investigating the delicate interplay between genetic susceptibility and environmental factors in ALD development could revolutionize risk assessment and treatment personalization. By integrating comprehensive genomic studies with longitudinal clinical data, researchers may identify specific genetic variants intricately linked to ALD susceptibility. Second, advancements in imaging techniques could enhance the non-invasive assessment of ALD progression and severity. Innovative methods such as multiparametric magnetic resonance imaging and molecular imaging hold promise in providing valuable insights into tissue composition, inflammation, and fibrosis, potentially mitigating the necessity for invasive liver biopsies.

In conclusion, this study contributes to our understanding of ALD pathology. However, addressing its limitations and outlining the path forward for research is crucial to advancing our understanding and refining clinical strategies for managing this complex ailment.

Conclusion

In summary, we successfully established ALI models using HepG2 cells and male C57BL/6 mice, IVT and IVV. ALD was characterized by elevated HMGB1 expression and reduced SIRT1 expression, supported by their negative correlation. Functional experiments showed the potential of SIRT1 in modulating HMGB1 acetylation and translocation, thereby alleviating ALD. This study provides novel insights into ALD pathogenesis and establishes a theoretical framework for the exploration of novel clinical interventions.

Supplemental Information

Supplemental Information 1 Uncropped blots

Click here for additional data file.

Supplemental Information 2 Supplementary figures

Click here for additional data file.

Supplemental Information 3 ARRIVE checklist

Click here for additional data file.

Additional Information and Declarations

Competing Interests

Author Contributions

Human Ethics

Animal Ethics

Data Availability

The authors declare there are no competing interests.

Juan Fu conceived and designed the experiments, analyzed the data, prepared figures and/or tables, and approved the final draft.

Wei Deng conceived and designed the experiments, authored or reviewed drafts of the article, and approved the final draft.

Jun Ge conceived and designed the experiments, prepared figures and/or tables, and approved the final draft.

Shengqi Fu performed the experiments, prepared figures and/or tables, and approved the final draft.

Panpan Li performed the experiments, authored or reviewed drafts of the article, and approved the final draft.

Huazhi Wu performed the experiments, authored or reviewed drafts of the article, and approved the final draft.

Jiao Wang analyzed the data, prepared figures and/or tables, and approved the final draft.

Yi Gao analyzed the data, authored or reviewed drafts of the article, and approved the final draft.

Hui Gao analyzed the data, authored or reviewed drafts of the article, and approved the final draft.

Tao Wu analyzed the data, prepared figures and/or tables, and approved the final draft.

The following information was supplied relating to ethical approvals (i.e., approving body and any reference numbers):

Hainan General Hospital Ethics Committee

The following information was supplied relating to ethical approvals (i.e., approving body and any reference numbers):

Hainan General Hospital Ethics Committee

The following information was supplied regarding data availability:

The raw data is available at Figshare: Fu, Juan (2023). Raw data.zip. figshare. Journal contribution. https://doi.org/10.6084/m9.figshare.23607297.v1.

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
