# Peer review of "Sirtuin 1 alleviates alcoholic liver disease by inhibiting HMGB1 acetylation and translocation"

_PeerJ, doi:10.7717/peerj.16480_

## Round 0.1 · original submission · Major Revisions

The three reviewers have given very specific and pertinent comments, please reply to their comments one by one.

**Language Note:** The review process has identified that the English language must be improved. PeerJ can provide language editing services - please contact us at copyediting@peerj.com for pricing (be sure to provide your manuscript number and title). Alternatively, you should make your own arrangements to improve the language quality and provide details in your response letter. – PeerJ Staff

Reviewer 1 ·

Basic reporting

no comment

Experimental design

no comment

Validity of the findings

no comment

Additional comments

Is there a direct interaction between Sirt1 and HMGB1?
What is the specific acetylation site on HMGB1?

Reviewer 2 ·

Basic reporting

1. The authors of this study aimed to investigate the role of high mobility group box 1 (HMGB1) in alcoholic liver disease (ALD), specifically by regulating SIRT1. These findings provide novel insights into the pathogenesis of ALD and offer a theoretical basis for the development of new clinical drugs. Overall, the research intent and logic of the manuscript are clear, but the English expression needs to be improved and needs to be proofread by fluent English-speaking professionals.
2. The citation of references in the manuscript is reasonable. There are still some studies reporting the role of HMGB1 in alcoholic liver disease, please elaborate on it in the Introduction. Such as Arch Physiol Biochem. 2021 Sep 20;1-9.
3. The structure of the manuscript conforms to an acceptable format of standard sections. The resolution and font size of the Figures need to be consistent. The Raw data appears to be complete and appropriate.
4. Because there has been a lot of research on HMGB1 and SIRT1 on ALD, the author needs to provide a detailed explanation in the abstract why these two factors are studied in ALD of the present study.
5. “Gao-Binge model was adopted to create the ALD model of mice” should be added References to support this method.
6. The study primarily investigates the immediate effects of SIRT1 activation on HMGB1 acetylation and translocation and the associated amelioration of ALD. However, the long-term effects on disease progression, recurrence, and overall patient outcomes are not addressed. In addition, the study may have a limited sample size and variability, especially in terms of patient samples. Larger and more diverse cohorts would provide more robust and representative results.
7. Zeng et al., have demonstrated that Inhibition of HMGB1 release via salvianolic acid B-mediated SIRT1 up-regulation protects rats against non-alcoholic fatty liver disease (Sci Rep. 2015 Nov 3;5:16013.). The author needs to clarify the differences between this study and the present study.
8. The significance of this paper is not expound sufficiently. The author need to highlight this paper's innovative contributions.

Experimental design

.

Validity of the findings

.

Reviewer 3 ·

Basic reporting

1.The overall presentation of the manuscript is professional, with standardized chapters, fluent language, and clear ideas.

2.The citation of references is reasonable and effective. Most of the references are within the last 5 years.

3.It is recommended to use consistent abbreviations throughout the manuscript, and provide the full word explanations when using an abbreviation for the first time. For example: SIRT1, ROS, PVDF.

4.The data and figures are basically clear. Figure 5A lacks scale labeled.

Experimental design

5.The background introduction can be further expanded to provide more information about HMGB1 and SIRT1, as well as their roles in alcoholic liver disease. This will help readers who are not familiar with the topic to better understand the significance of your research.

6.The study successfully established ALI models using HepG2 cells and C57BL/6 mice. ALD was characterized by up-regulated expression of HMGB1 and down-regulated expression of SIRT1, with a negative association between them. Functional experiments revealed that SIRT1 may regulate the acetylation and translocation of HMGB1, leading to the alleviation of ALD. The experimental design is clear, but the author needs to explain the difference or innovation between this study and other studies.

7.“HepG2 cells were digested by trypsin in logarithmic growth phase, thereby preparing single cell suspension, followed by seeding the cells in 96-well culture plates with the density of 4 × 103 (200 μL)/well. 24 h later, 10 μL of CCK-8 solution (Dojindo Molecular Technologies, Inc., Kumamoto, Japan) was filled to the well.” Please check this statement. Does 24h mean 24h after cell seeding? Or 24 hours after treatment?

Validity of the findings

8.Gao-Binge model should be supported by citing relevant literature to increase its credibility.

9.The explanation regarding figures 5A and 5B should be more detailed to help readers better understand the pathological features of this disease model.

10.While the discussion addresses the relation between ALD and oxidative stress, inflammatory reaction, lipid metabolism disorder, and other factors, it would be helpful to provide more context and cite specific studies or evidence to support these statements.

11.The discussion section should also include explanations of the study limitations and propose potential directions for future research.

---

## Round 0.2 · Minor Revisions

I will kindly request the authors of the need to address the following points in their revision:

1. Include detailed information on the number of replicates studied for each figure to ensure transparency and strengthen the statistical significance of the findings.

2. It is important to provide evidence of the purity of the nuclear and cytosol fractions isolated using the commercial kit. The authors should include data showing the enrichment of a nuclear marker and a cytosol marker in these fractions to assess any potential cross-contamination.

Reviewer 1 ·

Basic reporting

The author has made good revisions without any further comments.

Experimental design

The author has made good revisions without any further comments.

Validity of the findings

The author has made good revisions without any further comments.

Additional comments

Consider accepting the manuscript.

Reviewer 2 ·

Basic reporting

The author has considered my comments and made revisions, which I believe are sufficient.

Experimental design

After considering the comments, I can agree with his response and revisions.

Validity of the findings

After considering the comments, I can agree with his response and revisions.

Additional comments

The current manuscript can be considered for acceptance.

Reviewer 3 ·

Basic reporting

The author basically solved my concerns, and the quality of the manuscript was improved.

Experimental design

No more comments.

Validity of the findings

No more comments.

Additional comments

No more comments.

---

## Round 0.3 · accepted · Accept

Thank you for your responses to our concerns. Since these issues were not raised by external reviewers, this manuscript does not need to be sent for external review. Thank you very much for the opportunity to evaluate your research, and I hope to see your submissions again in the future.